# Control and Measurement Systems Supporting the Production of Haylage in Baler-Wrapper Machines

**DOI:** 10.3390/s23062992

**Published:** 2023-03-09

**Authors:** Michał Zawada, Mateusz Nijak, Jarosław Mac, Jan Szczepaniak, Stanisław Legutko, Julia Gościańska-Łowińska, Sebastian Szymczyk, Michał Kaźmierczak, Mikołaj Zwierzyński, Jacek Wojciechowski, Tomasz Szulc, Roman Rogacki

**Affiliations:** 1Center of Agricultural and Food Technology, Łukasiewicz Research Network, Poznan Institute of Technology, 61-755 Poznan, Poland; mateusz.nijak@pit.lukasiewicz.gov.pl (M.N.); jaroslaw.mac@pit.lukasiewicz.gov.pl (J.M.); julia.goscianska@pit.lukasiewicz.gov.pl (J.G.-Ł.); sebastian.szymczyk@pit.lukasiewicz.gov.pl (S.S.); michal.kazmierczak@pit.lukasiewicz.gov.pl (M.K.); mikolaj.zwierzynski@pit.lukasiewicz.gov.pl (M.Z.); jacek.wojciechowski@pit.lukasiewicz.gov.pl (J.W.); tomasz.szulc@pit.lukasiewicz.gov.pl (T.S.); roman.rogacki@pit.lukasiewicz.gov.pl (R.R.); 2Faculty of Mechanical Engineering, Poznan University of Technology, 60-965 Poznan, Poland; stanislaw.legutko@put.poznan.pl; 3Faculty of Control, Robotics and Electrical Engineering, Poznan University of Technology, 60-965 Poznan, Poland

**Keywords:** agriculture, agricultural machinery, baler-wrapper, silage applicator, density control, swath measurement, precision agriculture, sustainable agriculture

## Abstract

Baler-wrappers are machines designed to produce high-quality forage, in accordance with the requirements of sustainable agriculture. Their complicated structure, and significant loads occurring during operation, prompted the creation of systems for controlling the machines’ processes and measuring the most important work parameters, in this work. The compaction control system is based on a signal from the force sensors. It allows for detection differences in the compression of the bale and additionally protects against overload. The method of measuring the swath size, with the use of a 3D camera, was presented. Scanning the surface and travelled distance allows for estimating the volume of the collected material—making it possible to create yield maps (precision farming). It is also used to vary the dosage of ensilage agents, that control the fodder formation process, in relation to the moisture and temperature of the material. The paper also deals with the issue of measuring the weight of the bales—securing the machine against overload and collecting data for planning the bales’ transport. The machine, equipped with the above-mentioned systems, allows for safer and more efficient work, and provides information about the state of the crop in relation to a geographical position, which allows for further inferences.

## 1. Introduction

In this article, issues connected with the operation of fixed-chamber baler-wrappers, equipped with a work parameter control system, including precision agriculture systems, are examined. The main goal of this type of machine is to achieve high-quality forage, through the automation and adaptability of control systems to various environmental conditions. The baler-wrappers are machines used to form and ensilage forage as cylindrical bales. They combine the two functionalities of a round baler and a wrapper. This allows an increase in the efficiency, speeding up the haylage production process and improving quality, by reducing the oxidation processes, compared to the traditional production method [1]. Currently, there are several models of baler-wrapper available on the market. Based on the technical and commercial company literature, the most interesting of them are detailed here, focusing on the specific features of the presented equipment. The French company, Kuhn, offers three series of machines of this type. They differ in terms of equipment, but also in their functionality. Particularly noteworthy is the FBP 3135 series baler-wrapper, which, as an option, has an integrated bale weight measurement system. The machine is equipped with an automatic arm, that adjusts the position in relation to the bale during wrapping, obtaining the optimal foil coverage in terms of strength. The method of applying the foil has been programmed in such a way that the machine wraps the bales in the places most exposed to damage—meaning on the cylindrical surface [2]. Another example of a baler-wrapper that uses technologies in its work, different to those presented above, is the Fusion 3 series, by McHale. It is equipped with an 18 cylinders bale chamber and a vertical wrapping ring. Its unique feature is its system of transferring the bale from the chamber to the wrapping module, using a mechanism minimizing the number of drives. Additionally, the operator can control and change the machine’s work parameters, using the console placed in the tractor cabin [3].

Optimizing the functionality of machines for creating haylage is an issue raised by manufacturers and scientific units on the global market. There are very novel and innovative approaches, such as a prototype combining the functionality of a belt rake with a baler-wrapper, designed by a team of scientists [4]. There are many baler-wrappers available on the agricultural machinery market; however, most of them do not have systems that are compatible with precision farming. There are known publications concerning the placement of atomizer dosing ensilage agents above the pick-up [5], but variable fluid dispensing is not included there. An interesting solution is also the development of press technology with its own drives. The Vermeer ZR5-1200 press is a commercial example of this [6].

Meeting the requirements of precision farming, prompts the search for measurement methods that allow an analysis of the size of crops in the fields. Numerous publications show the significant importance of 3D sensors in applications in agricultural machines, due to the provision of a large amount of precise data, allowing for the measurement of e.g., volume [7]. There are known solutions that, using 3D sensors, achieve centimeter measurement accuracy in geomorphological measurements (SFM and MVS) [8,9]. Most of the methods mentioned above require the use of expensive computational units to analyze the results. There are known 3D measurement methods operating on the principle of the reconstruction of scanned elements, by combining multispectral images with point clouds, using the reflection coefficient [10]. Methods based on the analysis of a stereoscopic image are also used to measure the yield [9,11,12]. The additional use of camera tilt compensation in stereoscopy allows a measurement accuracy of about 4 cm to be obtained; however, an appropriate computing unit is required, to implement the algorithm [11]. In addition, yield measurement can be carried out using laser sensors, ultrasonic sensors, and high-resolution radar images [9,13]. There is a noticeable growing interest in systems supporting the work of presses and baler-wrappers. These include, among others, the driver assistance system, for detecting and measuring the swath; the scanner is based on the signal from a LIDAR sensor. According to the manufacturer, the system enables the measurement of the size of the swath and the automatic directing of the tractor to the swath, with the driving speed adjusted to the size of the material collected [14]. Another element supporting the production of forage is humidity sensors, which, according to the manufacturer (Kuhn), have a measuring range of up to 40% of water content. Since 2021, the described machines have also been equipped with systems for weighing the produced bales [15,16]. 

The aforementioned quality of the forage produced by the machine, results from the density of the bales. There are known solutions that, based on the pressure measurement in the hydraulic system, draw conclusions about the density of the bales being pressed. It is necessary to filter the signal from pressure sensors, because it is characterized by large fluctuations [17]. Automation of the net feeding process, based on the analysis of the bale density signal, allows an increase in productivity of 14%, by controlling the pressure in the baler’s hydraulic system. The authors indicate the need to develop appropriate signal filtration. In the tests, the crushing force was compared with the indications from the tractor’s PTO and the number of bales produced was counted [18].

The bale weight measurement is based on the coupling of mechanisms with force sensors on the receiver, that transfers bales from the pressing chamber [17]. Systems of applying additives, supporting the creation of high-quality fodder in balers and baler-wrappers, in the form of nozzles dosing ensilage agents mounted above the pick-up of the machine, are also observed [19]. The significance of uniform dosing of the agents mentioned above, in order to obtain high-quality fodder, is shown to be of high importance, which directly indicates the importance of measuring the yield [20,21,22,23]. The conclusion about the amount of silage agent applied is based on: moisture content, length of plant material (based on the length of the road traveled by the tractor), and bale pressing pressure [19].

The aforementioned examples of machines and methods, have control and measurement elements used to determine the yield—especially important for sustainable farming and precision farming requirements. Farmers’ desire to improve soil fertility has consequences that adversely affect the environment if the relevant rules are not followed. One of the most popular courses to improve soil efficiency and increase yields is the use of artificial fertilizers and plant protection agents. These are an inseparable part of agriculture focused on maximum production and the highest profit [24]. Their excessive use may lead to the release of nitrogen and phosphorus compounds into groundwater and surface waters, which may result in the extinction of organisms living in the water reservoirs, and even their release into drinking water. This is especially dangerous for children and people who use well-water resources [25,26]. Limiting the above-mentioned negative effects around the world, drives the search for methods to reduce environmental contamination. This had led to the creation and development of a sustainable agriculture system, the main assumption of which is to reduce the impact of crops on non-agricultural areas and reduce the amount of waste unused in production processes, and to limit the use of chemicals [27,28,29]. Its characteristic features are the aspiration to reduce production on industrial farms, the protection of soil and biological resources, by organizing agricultural production in a way that does not cause negative changes to the natural environment, or making minor changes to reduce negative phenomena, such as erosion. Sustainable agriculture is a management system that combines economic, social, and ethical aspects, with ecological safety [24,30,31]. An example of combining environmentally friendly methods with modern technologies of automation are baler-wrappers, and the equipment dedicated to supporting their functionality. It can therefore be concluded that the external factors mentioned above, will influence the further development of methods to support more ecological crops and fodder.

In this publication, the focus is on aspects connected with bale density control, and the original method of estimating the volume of the collected swath; also, a system of variable dosage of agents regulating the process of ensilage forage, is presented. The paper has an interdisciplinary character, due to the combination of many branches such as mechanical engineering, automation, electronics and, above all, agriculture.

The issue of a systemic approach to the creation of high-quality forage is part of a project implemented by the Łukasiewicz Research Network–Poznan Institute of Technology, at the Center of Agricultural and Food Technology. The publication is the result of work on the project no. POIR.04.01.04-00-0067/18, entitled “Baler and baler-wrapper for harvesting roughage into cylindrical bales with a system for monitoring and influencing the process of their creation”. Taking into account the current trends in agriculture, including the pursuit of environmental protection and compliance with the requirements of sustainable agriculture [24,32], the presented baler-wrapper systems are important from the point of view of the end users. Progressive agriculture should be based on specialist knowledge, and open to new technologies that will improve the profitability of the farm production and reduce the negative impact on the environment. One possible way to achieve this is to apply the elements of precision farming to machines [33,34,35]. This article presents a method of creating a field yield map based on the estimation volume of the pressed crop. On this basis, it is possible to develop an application map of fertilizers, enabling their variable dosing to increase fertility. The following hypothesis was formulated: the use of modern precision farming systems allows the control of the process of creating bales and regulating the forage ensilage.

## 2. Materials and Methods

### 2.1. Subject of Research

The importance and specificity of supporting the production of haylage, presented in the introduction to the article, contributed to the design of systems adapting to the current working conditions of the prototype of the baler-wrapper, equipped with several innovative solutions. The recording of key parameters and their analysis, is in line with the guidelines set by precision farming. The published research was carried out during functional tests of the prototype of the fixed chamber baler-wrapper in October 2021. The machine was implemented within the POIR.04.01.04-00-0067/18 project, by the Łukasiewicz Research Network–Poznan Institute of Technology, and the factory of agricultural machinery Metal-Fach Ltd. The research had an experimental character and was carried out in terms of strength tests and verification of the correct operation of the bale control systems, which is the subject of the publication.

### 2.2. Input Conditions of the Research

The production of forage, by pressing into cylindrical bales and wrapping with foil, is largely dependent on external weather conditions. Machine tests were carried out on a 17 ha meadow, in autumn, under favourable weather conditions: the average temperature was 15 °C, and, no rainfall during the previous four days resulted in the harvested material having an average humidity of about 50%. The measurement was made based on a series of repetitions with the Dramiński HMM hygrometer, intended for measuring the humidity of the pressed bales. The measurements were performed at three measurement points on each bale (in the geometrical center of the bale, at half of the bale’s radius, and the extreme position). The measurements were carried out on a series of 10 randomly selected bales. As shown in the literature [36], it is possible to obtain high-quality silage even in the case of not adding ensilage additives in the form of inoculants, using the ensilage method, in the form of pressing and wrapping the bales with foil in conditions of high humidity. The tests were made to relate the operation of the machine systems to the characteristics of the collected material, which affects the bale wrapping process and the uniformity of the bale formation.

### 2.3. Haylage Production Support Systems

An important feature of the baler-wrapper is the modularity of the proposed systems. There is the possibility of supporting the work of the machine with a wide range of amenities, assuming the possibility of switching them on depending on the need and adapting to the conditions prevailing in the field. The most important systems supporting the production of haylage include (Figure 1):the bale compression ratio control system;the swath size recording system;the system of variable dosing of ensilage additives;the bale weight control system (not described in detail in this article).

The article describes more extensively selected additional modules of the baler-wrapper.

### 2.4. Bale Compression Control System

The ratio of bale compression is measured using electronic force sensors. In the beginning, there was also some research carried out with the hydraulic equivalent of the measuring system; however, due to the failure rate and special working conditions, the concept was abandoned. The force measurements are made at two points, on opposite sides of the machine, to detect differences in the bale density on both sides. Force sensors are mounted on one side on the stationary part of the machine, and on the other side they are attached to hooks, that block the possibility of opening the chamber where the bale is compressed. The bale, which increases in volume, pushes against the flap, and the latter, through the hooks, affects the force sensors.

There are six stages in the method of controlling the uniformity of the compression ratio of the bale (Figure 2). The principle of operation of the system should be considered in relation to the tractor track, or actually to the side of the pick-up, where the swath is collected (Figure 3).

Stages of the bale compressing (Figure 2, Figure 3):

Stage 1: The force sensors’ indications are tared when driving begins. They measure low values of forces while driving, resulting from disturbances, e.g., uneven terrain. The operator changes the tractor’s trajectory by driving for some time on the left side of the swath, and then the same on the right side (avoiding slalom driving), until the moment when the forces for the left and right sides are building up.

Stage 2: After observing the increase in the forces, the operator continues driving, in order to obtain a difference between them of about 500 N, and changes the driving track.

Stage 3: After changing the driving track to allow material to be picked up from the opposite side of the pick-up, the situation is reversed. The force from the opposite side has a higher value.

Stage 4: Another change in the driving track, in order to equalize the measured forces and achieve the assumed bale density.

Stage 5: The user decides to stop the machine and start wrapping the bale with a net. The measured forces decrease due to the continuous rotation of the bale inside the baling chamber (the last stage of compressing the bale).

Stage 6: After wrapping with the net, the bale is transferred to the foil wrapping unit. The forces are reset when the press chamber is opened. The user waits for information from the controller when the baling chamber is closed and starts the whole procedure anew. After wrapping with foil, the bale is ejected from the film wrapping module.

### 2.5. Swath Size Recording System

The presented method of measuring the swath volume is one of the main innovations used in the machine. The current swath size information is processed by the controller, to create field yield maps and for controlling other systems, e.g., for the variable dosing of ensiling agents. The measurement is made using the 3D O3M261 sensor by IFM (receiver), intended for use in mobile machines, and a dedicated O3M960 infrared illuminator (transmitter) (Figure 4). The devices are connected to each other by a cable that synchronizes their operation, consisting in measuring the time difference between the emitted infrared beam from the transmitter, and the time of its reflection from the obstacle and return to the receiver. The signal is interpreted by the device driver as a distance. To ensure proper operation, as well as to increase the accuracy of the readings, a dedicated, adjustable handle was made for the tractor, with the possibility of changing the height of the position of the measuring devices in relation to the ground and adjusting the inclination angle of the 3D sensor. The typical measurement accuracy of the sensor used is 5–10 cm, for a measuring distance of less than 5 m between the sensor and the detected element. Measurement accuracy also depends on the size of the detected object and the viewing angle of the 3D camera.

The measuring set is directed at a specific angle, α, at the height, *h*, in relation to the ground. The devices were pre-calibrated in the dedicated vision assistant software provided by the manufacturer, which allows the user to check the read values and compare them with the actual height, *h*. The 3D sensor enables the measurement of a 3D image with a resolution of 64 × 16 pixels, while the actual viewing angle is 95 × 32 degrees. The pixels form a point cloud (Figure 5), which can be divided into the ROI (regions of interest)—varying the settings makes it possible to concentrate the measurements in an interesting range and to average the measurements within each group. The actual field of view of the 3D sensor is marked in blue (Figure 6). As a result, there is no need to analyze each pixel individually. The main controller analyzes the pre-processed data.

The measurement method of estimating the total volume of the swath, in general, comes to adding up the volumes of the averaged cuboids described on the swath, based on the distance travelled by the tractor in the assumed time, and on the basis of the averaged cross-sectional area of the swath in the assumed time, t. The data is saved in the controller’s memory, with a specified frequency resulting from the parameter t.
(1)$$V_{n}=\sum_{i=1}^{n}V_{i}\left[m^{3}\right],$$
where:
$V_{n}—\mathrm{total}\;\mathrm{volume}\;\left[m^{3}\right],$$V_{i}—\mathrm{single}\;\mathrm{volume}\;\mathrm{measurement}\;\left[m^{3}\right].$

The method of measuring the volume depends on the controller cycle time, $t_{c}$, and the time, t, assumed by users, and it is interpreted as a single cuboid escribed on the swath (volume measurement). For this purpose, the parameter of the number of controller cycles, z, in the assumed time, t, is determined.
(2)$$z=\frac{t}{t_{c}},$$
where:
z—is number of controller cycles in the assumed time,$t—\mathrm{is}\;\mathrm{the}\;\mathrm{assumed}\;\mathrm{time}\;\mathrm{of}\;a\;\mathrm{single}\;\mathrm{volume}\;\mathrm{measurement}\;\left[s\right],$$t_{c}—\mathrm{is}\;\mathrm{the}\;\mathrm{controller}\;\mathrm{cycle}\;\mathrm{time}\;\left[s\right].$

After the appropriate parameterization of the 3D sensor, it is possible to measure the height of each ROI, and thus calculate the area. Each ROI has a defined fixed width. The diagram below (Figure 7) shows the method for measuring the swath volume, $V_{i}$, at time t. The number of bar graphs corresponds to the z parameter, i.e., the number of controller cycles at time t. The horizontal axis of the graph shows the single ROI recorded for each cycle, whereas the vertical axis is the average height of each ROI, directly read from the 3D sensor—this is possible thanks to appropriate parameterization.

The next step is to estimate the height of the described cuboid. For this purpose, the method of calculating the distance between the recorded geographic coordinates, which are saved in accordance with the main controller cycle, was used. Due to the registration of small sections, a simplified method of calculating the distance between successive geographic coordinates was used, according to the following formulas and relationships [25,37]:(3)$$\Delta \emptyset =\emptyset_{2}-\emptyset_{1},$$
(4)$$\Delta l\;=l_{2}-l_{1},$$
(5)$$\emptyset_{m}=\frac{\emptyset_{1}+\emptyset_{2}}{2},$$
(6)$$D=R\sqrt{{\left(\Delta \emptyset \right)}^{2}+{\left(\cos\left(\emptyset_{m}\right)\Delta l\right)}^{2}},$$
where:
R—is the radius of the Earth,Δ∅—is the difference in latitude,Δl—is the difference in longitude,$\emptyset_{m}—\mathrm{is}\;\mathrm{the}\;\mathrm{mean}\;\mathrm{latitude}.$

The method of calculating the distance between the coordinates has been tested during the research. Despite the use of simplifications, the calculated distance has a satisfactory error that enables the mapping of the driving tracks on the field map.

The formula below (7) explains the method for estimating the swath volume. As mentioned before, the method amounts to describing cuboids on the swath and determining their volume. The first step is to calculate the flat area, by summing the single ROI $x_{k}$ for one cross-sectional area, and taking the average of the measurements. Then the procedure is repeated for the z-number of the swath sections, and the mean of all averaged areas is determined.
(7)$$V_{i}=\sum_{j=1}^{z}D_{j}\cdot \left(\frac{\sum_{j=1}^{z}\frac{\sum_{k=1}^{n}x_{k}}{n}\left[m^{2}\right]}{z}\right),$$
where:
$D_{j}—\mathrm{calculated}\;\mathrm{distance}\;\mathrm{based}\;\mathrm{on}\;\mathrm{coordinates}\;\left[m\right],$t—$\mathrm{assumed}\;\mathrm{time}\;\mathrm{of}\;a\;\mathrm{single}\;\mathrm{volume}\;\mathrm{measurement}\;\left[s\right],$$x_{k}—\mathrm{single}\;\mathrm{ROI}’s\;\mathrm{surface}\;\mathrm{area}\;\left[m^{2}\right],$n—number of ROIs.

To determine the volume $V_{i}$, the averaged area of the cross-sections was multiplied with the length, estimated based on the distance between the geographic coordinates. The last step is summing up the individual volumes with the cycle determined by time t.

### 2.6. System of the Variable Dosing of Ensilage Additives

The dosing of ensilage additives to forage is a well-known process used in agriculture. It aims to increase the quality of forage by the use of preservative agents, aiding the ensilage process and reducing the risk of losses associated with spoilage (mold, rot, etc.) [33]. The validity of using ensilage additives is tested in various centers around the world. The researchers show the need to use additives, in particular, for legume plants and species of grasses that are ensilaged at high temperatures and thus have too high a dry matter content in green fodder [33,36,38]. The variable dosing system is a component of the entire baler-wrapper control system and is managed from the machine’s main controller (Figure 8). It uses, among other things, the GPS module to retrieve information about the movement of the machine or a standstill. The ensiling agent is dosed automatically while driving and stops at standstill times, e.g., when wrapping the net.

The method of variable dosing of the ensilage agents, works based on the swath volume measurement from the 3D sensor (Figure 9). On this basis, it is possible to determine the approximate swath density coefficient, and to apply a dose based on the estimated weight, or directly based on the magnitude of changes in the registered volume. Most producers of ensiling agents provide very general information on the dosage of the product—a recommended volume of the ensilage agent mixed with water for the possible total weight of the green fodder (performance parameter). To calibrate the system, it is necessary to enter data on the maximum allowed pressure in the system and dosage, e.g., 10l/50t. The main goal is to save ensilage agent, but also to achieve the best forage quality by dosing more fluid in situations where it is needed. The landform of the meadow has a large influence on the harvesting efficiency. The system adjusts itself to the changing conditions in the field during the collecting of the swath. The further part of this article confirms the importance and the need to create such a system.

## 3. Results

### 3.1. Bale Forming Parameters

Bale compression control during the tests was carried out, with synchronous recording, as well as other correlated parameters of the baler-wrapper operation, i.e., the torque transmitted by the tractor PTO and the change in bale weight, using pressure sensors on the baler axle. Such readings, for analysis purposes, must be processed, since the recorded signal has many distortions that prevent their practical interpretation. To this end, during the initial analysis, the signals were finally processed with a low-pass Chebyshev filter, of 5 Hz. An example of the signal runs, their spectra, and the post-processed signal are shown in Figure 10. 

Taking into consideration the characteristics of the operation of the controllers used to operate the baler-wrapper machine, in the further stage of the analysis signal filtering was replaced by signal averaging, which is easy to implement in the controllers. The results obtained in this way are equivalent in practical applications.

Compression distortions arise, and, theoretically, it would be possible to directly use such a signal in a control system. However, taking into account the whole spectrum of possible disturbances during operation, it was assumed that all signals should be averaged in the same way. A summary of an example pattern of these parameters is shown in Figure 11. 

From the analysis of the recorded parameters, it can be determined that the value of the torque on the tractor’s PTO during idle reaches an average value of 50 Nm. It was assumed that the effective value of the torque during the operation of the machine is correspondingly smaller, which is taken into account in the subsequent runs. If we do not analyze the irregularity of the compaction of the bale over its width, it can be assumed that the measure of the compression is the sum of the forces acting on the flap (Figure 11a). The transverse dynamics of the machine, where the center of gravity is high (about 1.5 m relative to the ground), can cause large fluctuations in the compaction forces on both sides of the baler. This can be observed by analysing the patterns of pressure forces on the axle (Figure 11c). The design of the baler-wrapper makes it possible to assume that, during bale formation, almost the entire increase in the weight of the harvested material (Figure 11b) until the flap is opened and the bale is flipped onto the wrapper, is recorded by sensors on the axle. They are located in the area of the geometric center of the press chamber, and their shift of a value of 179 mm results in an underestimation of 5%, which requires the application of an appropriate correction factor. 

In the course of the bale formation, several characteristic points can be distinguished in the chamber of the baler-wrapper, repeating regularly in each cycle of operation (Figure 12). 

The bale is formed from the start of work run A to point D, at which point the decision is made to stop collecting the material. This is the peak point in terms of power consumption from the tractor’s PTO, and the maximum bale density occurs here. During the whole cycle between points A and D, point B can also be observed, where the most dynamic increase in torque demand occurs, and point C, where the bale reaches its nominal size and the compaction process begins. Between points C and D, a rapid increase in the compression force can be seen. At the CD and DE sections, a great similarity can be observed in the torque and compression force trends. In section DE, net wrapping of the bale occurs. Between points E and F, the chamber is opened and the bale is flipped onto the wrapper. It should be noted, that due to the modular design of the baler, the change in axle pressure at this stage can also serve as an additional measure of the bale weight (section E–F), which makes it possible to analytically determine the longitudinal balancing forces of the baler-wrapper. After the bale is flipped onto the wrapper, the recorded force increases theoretically by 51%, concerning the nominal bale weight; during the tests, the average increase in force was 40% (Table 1). Accepting this correlation, the weight of the bale can be determined by obtaining values ${\bar{F}}_{2}\left(EF\right)$, that differ by no more than 5% of the value read at point E ($\bar{F}\left(E\right)$). At point G, the bale is thrown into the field, completing the entire bale formation process.

Table 1 shows examples of the recorded values of the thrust, torque, and compression force.

The collected data show that bale weight is not a direct function of bale density, and must still depend on other factors during the bale formation—their mutual variation is shown in Figure 13. Based on the data presented, it can be concluded that density is a less predictable parameter than the bale weight and has a much larger scatter. For the data presented, the variation coefficient for the bale is 8.4%, and for the density is 20.2%.

### 3.2. Swath Efficiency Map

Figure 14 shows a swath efficiency map, created based on the results obtained by the author’s swath volume estimation method. A color scale has been used to show the distribution of the swath volume over the field area, dividing the results into eight groups about the maximum swath volume result. Geographical coordinates from the GPS receiver are associated, in the main controller, with the calculated volume.

### 3.3. Classification of Bales Made

Managing the machine’s modules from the main controller, in combination with the GPS signal, gives the possibility to generate many different variants of maps. One of them is the creation of a map of the distribution of bales in the field, with additional information, e.g., compression value (Figure 15), bale weight, numbering, moisture content, etc.

## 4. Discussion

### 4.1. Bale Forming Parameters

Obtaining information on the uniformity of the degree of density in the process of forming bales has a very significant impact on the nutritional quality of the forage obtained. 

In Figure 11 and Figure 12, the very similar characteristics of the changes in the torque recorded at the PTO and the forces recorded at the bale density control sensors can be seen. The extreme assumed by the operator compression values read from the force diagrams can be estimated at 5000 N, corresponding to a torque of approximately 450Nm. Therefore, it can be concluded from this that, by focusing only on the measurement of the moment, it is possible to estimate the bale compression force. The disadvantage of this type of solution is undoubtedly the lack of information on the difference in the values of the compression forces for opposite sides of the formed bale, only information on the overall, averaged value of the density is obtained. The presented measurement concept could also find application in conventional balers. 

The combination of the parameters of the weight of the bale created, compression force, and the value of the torque generated at the tractor’s PTO, also provides comprehensive information on the entire machine, and allows a warning system to be prepared for the operator, about the danger of overloading the machine. This is very important when making bales with a moisture content of more than 50%, whose final weight can reach up to 1 ton. The next step is to reference the physical conditions (weight, density, etc.) as a function of the inflicted dose of ensilage, which has reference to the quality of the feed.

### 4.2. Swath Efficiency Map

The developed swath efficiency map fits into the widely known technologies associated with precision agriculture. However, a different approach to the subject was used here, since information on local swath size was utilized using 3D cameras and the original measurement method (p.2.5). The map presented in Figure 14 shows the changes in swath size in the field in a scaled form of the results. The blue color indicates the smallest swath volume (up to 0.014 m^3^), while the red indicates the largest (range 0.227–0.416 m^3^). As a confirmation of the validity of the obtained results, the fact that the swath volume overlaps with the terrain—in places of local hills, where water runoff occurs during precipitation and storage in the soil is hindered—is evidenced by a lower efficiency being registered. 

Based on the obtained characteristics of the swath size distribution, it is possible to create an application map for variable dosages of fertilizers, or to draw up an irrigation plan for areas where efficiency could be higher. The approach used makes it possible to save the time and expense of conducting soil tests and detailed analyses of the chemical composition of the entire field. 

### 4.3. Classification of the Bales Made

The presented map of bale numbering and distribution, with selected physical parameters of the baling and wrapping process (Figure 15), is one of the variations that can be generated. A map of this type provides an opportunity to plan the logistical processes associated with the transportation of bales to the farm and includes information on the total number of bales produced. 

The presented solution allows the planning of logistics associated with transport, among other things. Information on the size and uniformity of the bales’ compression makes it possible to predict how they will be placed on the trailer, the number of trips, and inventory-related planning. Bales with a lower degree of compression are susceptible to damage during handling, and require, for example, less dynamic loader movements. With the aforementioned information, it is possible to adjust the number of trips and the dynamics of the loader to the physical parameters of the bales. 

The main innovation that distinguishes the described mapping system, is the ability to identify which area individual bales are from and relate this information directly to the quality of the feed, which has a significant impact on the propensity of livestock to consume it without problems [29,30]. 

## 5. Conclusions

Elements of precision agriculture are being used in a widening spectrum of agricultural technology machinery. This includes balers, which are used in the production of high-quality feed. It should also be mentioned in this regard that they are relatively complex machines, requiring the use of support systems and production control. This article describes selected automation topics that improve the efficiency, productivity, operational reliability, and ergonomics from the operator’s point of view, and, most importantly, control systems that affect the quality of feed for livestock feeding. 

A key element for the operation of most of the project’s innovations was the development of a method for estimating the volume of swaths raked into rows. The word estimation is deliberately used, because the nature of the harvested material varies and depends on many variables, such as the type of rake used, the volume, type, and humidity of the crop, etc. It is important to determine the number of changes and characteristics of the forage material for adjusting ensilage rates and to determine the varying distribution of the amount of material on the field surface, for analysing the total efficiency. The subsequent step will be to establish a coefficient defining the relation between the volume and weight of the bale, which will enable the development and determination of simplifications in the measurement system. A major innovation in the implementation of the project is the generation of a field performance chart, which is important in farm logistics management. The identification of the size of the swath, combined with the machine’s positioning signal, makes it possible to make an application map for the variable application of fertilizers in the field. Considering economic and environmental aspects, these are some of the key elements of the project, forcing the need for precision farming systems. 

The hypothesis stated in the introduction confirms that the use of modern precision farming systems allows for the control of the process of creating bales and regulating the forage ensiling process, and has been proven with the example of the method of density control, the concept of dosing ensiling agents, and recording the weight and volume of the swath.

The presented baler support systems are part of patent applications, due to their high degree of innovation.

## Figures and Tables

**Figure 1 sensors-23-02992-f001:**
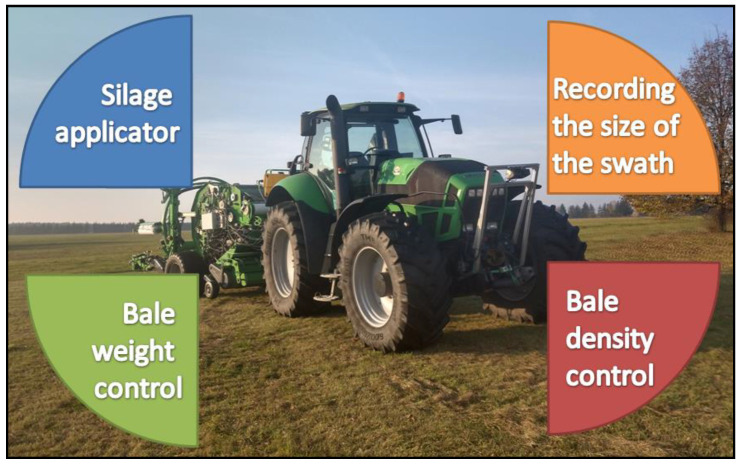
Baler-wrapper control systems.

**Figure 2 sensors-23-02992-f002:**
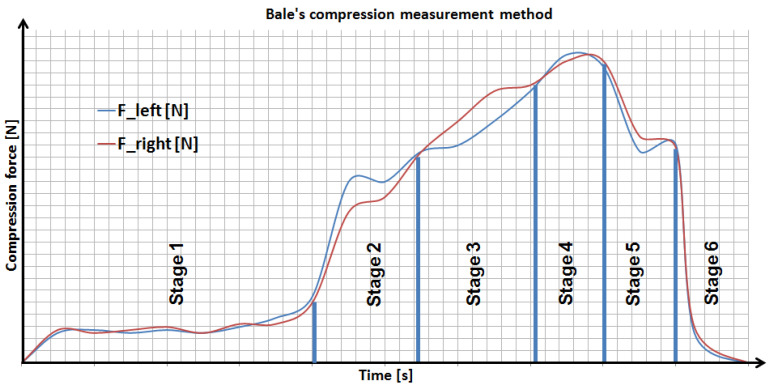
The bale compression measurement method.

**Figure 3 sensors-23-02992-f003:**
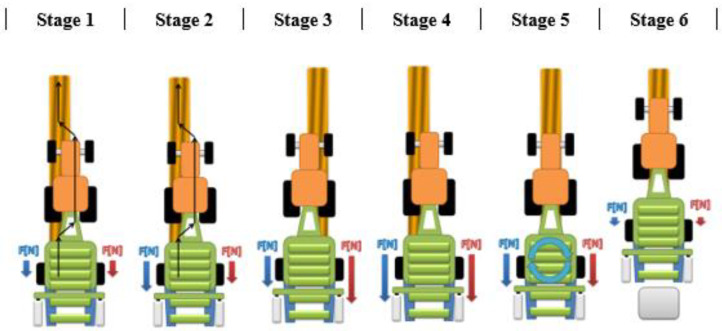
The relation between the forces of pressing the bale and the tractor driving trajectory—differences in force increases depend on the place where the swath is taken by the machine.

**Figure 4 sensors-23-02992-f004:**
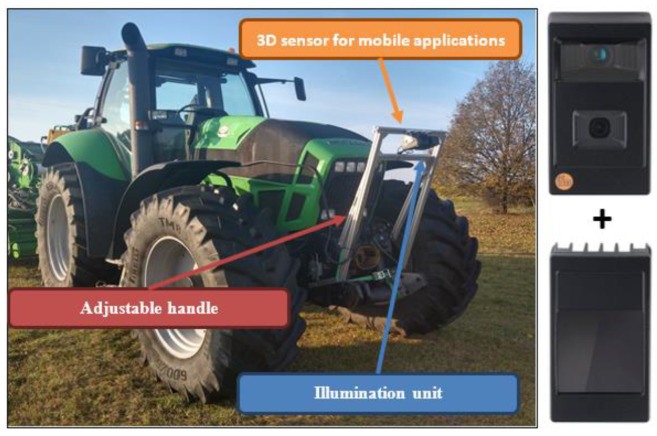
The swath size measuring system consisted of a 3D sensor, illuminator (made by IFM electronic), and adjustable handle.

**Figure 5 sensors-23-02992-f005:**
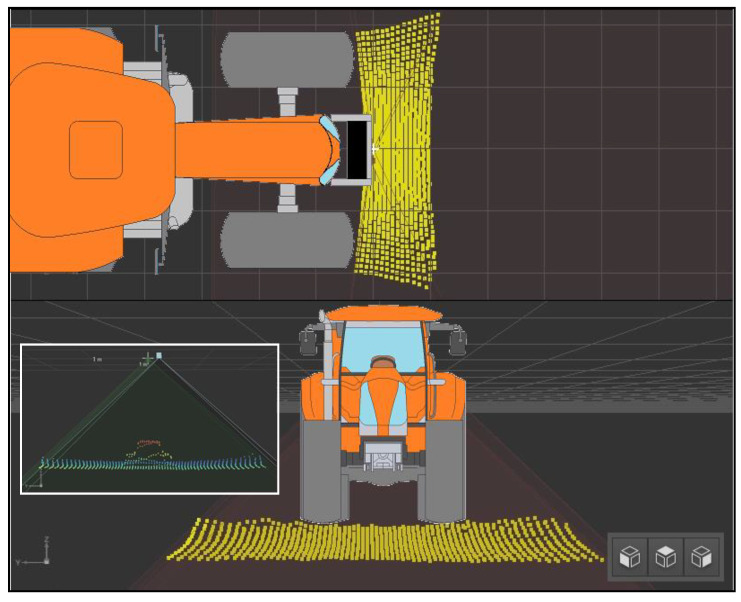
View of the cloud of 3D sensor measurement points read from the vision assistant program.

**Figure 6 sensors-23-02992-f006:**
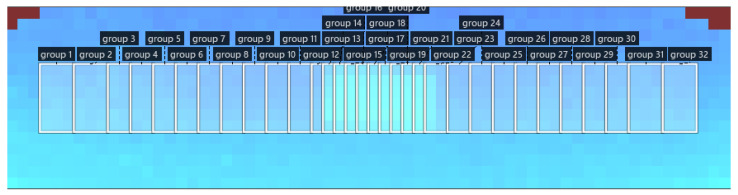
ROI (regions of interest) in swath size measuring system.

**Figure 7 sensors-23-02992-f007:**
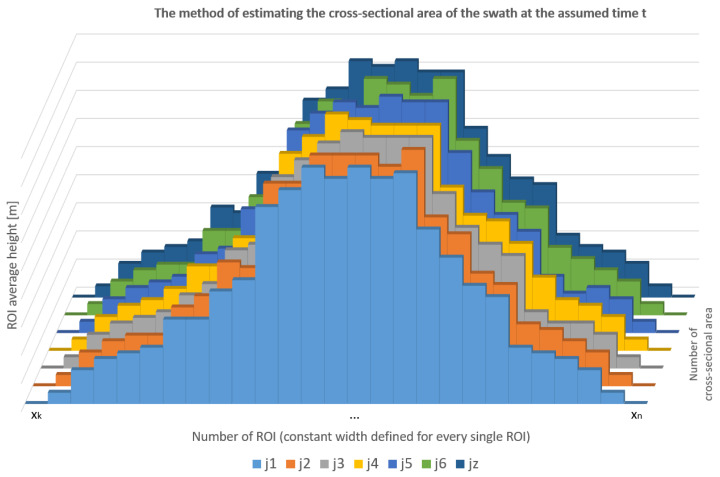
The method of estimating the cross-sectional area of the swath at the assumed time, t.

**Figure 8 sensors-23-02992-f008:**
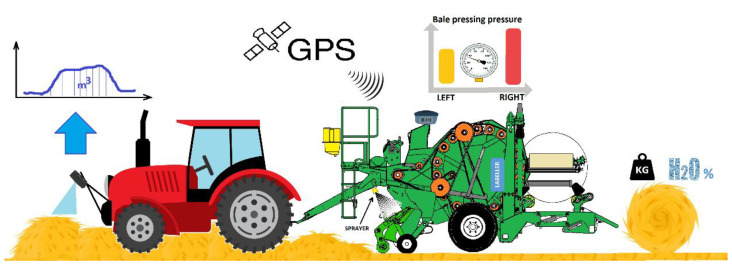
Silage support systems in the baler-wrapper.

**Figure 9 sensors-23-02992-f009:**
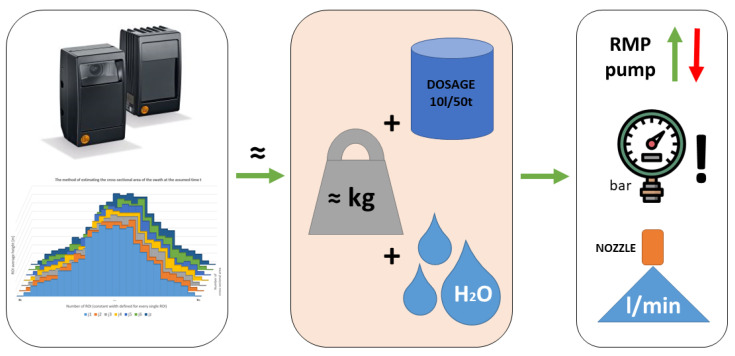
Schematic diagram of the system of the variable dosing of silage.

**Figure 10 sensors-23-02992-f010:**
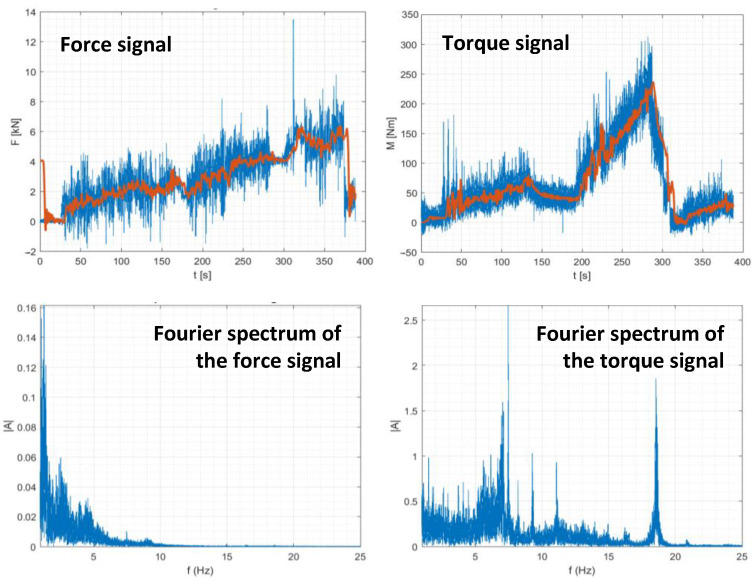
Characteristics of force and torque measurement signals.

**Figure 11 sensors-23-02992-f011:**
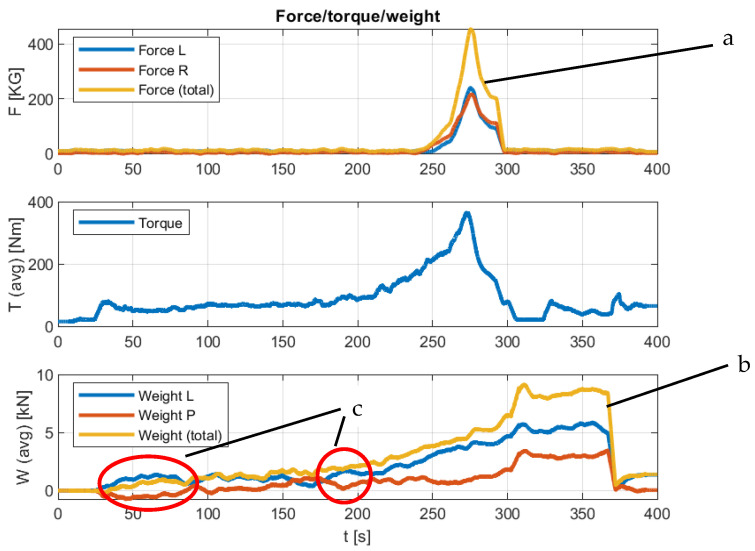
Summary of averaged values of runs: (**a**) total compression force; (**b**) total axle load; (**c**) symmetrical fluctuations of the pressure force.

**Figure 12 sensors-23-02992-f012:**
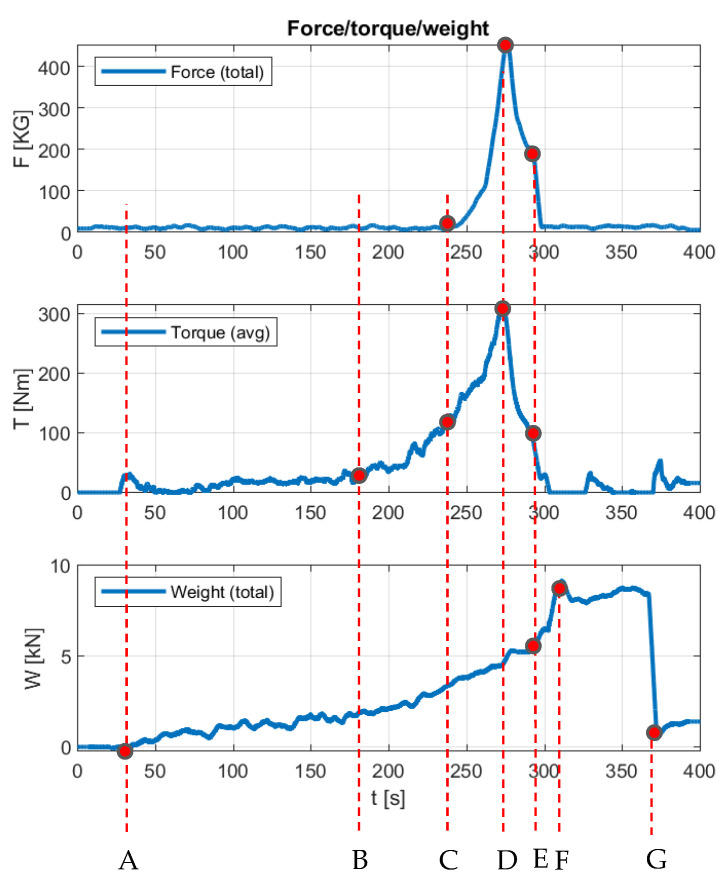
An example of the process of creating a bale, with the characteristic points of the baler-wrapper cycle:(**A**–**G**)—characteristic stages of machine operation.

**Figure 13 sensors-23-02992-f013:**
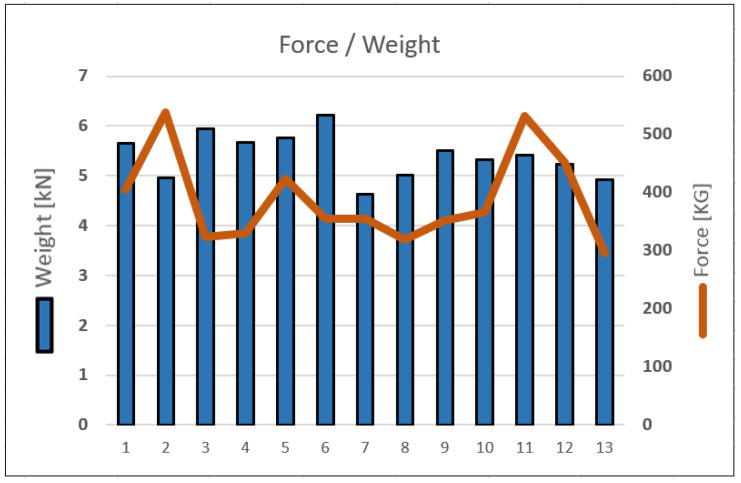
Dependence of bale weight and compaction parameters.

**Figure 14 sensors-23-02992-f014:**
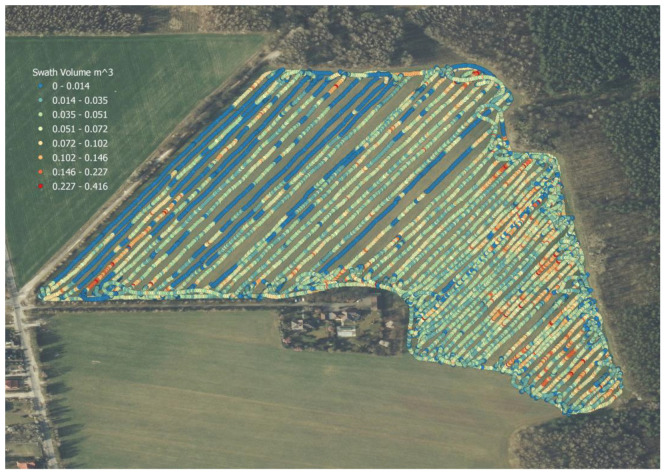
Swath efficiency map, prepared with the proprietary method of swath volume estimation.

**Figure 15 sensors-23-02992-f015:**
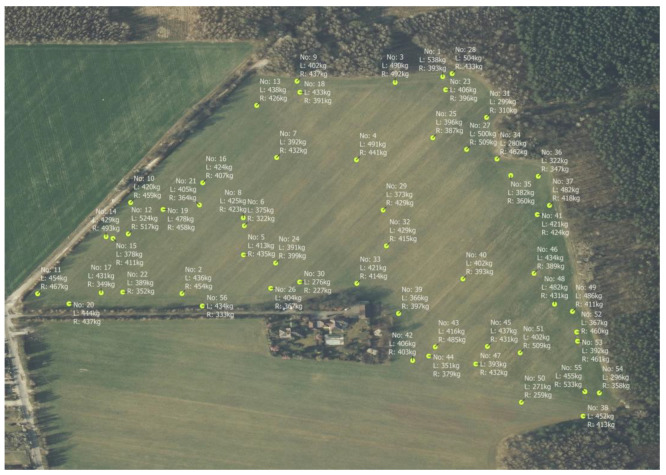
An example of a bale distribution map, with information on maximum compaction.

**Table 1 sensors-23-02992-t001:** A summary of the measurements averaged over the parameters of the baler, concerning the characteristic operating points of the machine as described in Figure 12, where: $\bar{F}$—is the average value of the forces, $\bar{T}$—is the average value of the torque generated at the tractor PTO, and $\bar{Z}$ —is the average value of the maximal bale density (the average of the maxima read from the density sensors from both sides of the machine).

No.	$\bar{F}\left(E\right)\left[kN\right]$	$\bar{F}\left(F\right)\left[kN\right]$	$\Delta \bar{F}\left(EF\right)\left[\%\right]$	${\bar{F}}_{2}\left(EF\right)\left[kN\right]$	$\bar{T}\left(D\right)\left[Nm\right]$	$\bar{Z}\left(D\right)\left[kG\right]$
1	5.65	8.79	48	5.48	180	404
2	4.95	8.44	71	5.26	232	537
3	5.95	9.87	66	6.15	172	323
4	5.67	9.39	66	5.85	251	330
5	5.76	8.79	53	5.48	260	423
6	6.22	9.40	51	5.86	251	355
7	4.63	7.62	65	4.75	253	354
8	5.01	8.43	68	5.26	245	318
9	5.51	9.05	64	5.64	276	351
10	5.33	8.15	53	5.08	253	365
11	5.42	8.69	60	5.42	299	531
12	5.23	8.14	56	5.07	314	451
13	4.93	8.15	65	5.08	259	295

## Data Availability

Not applicable.

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
