# Peer review of "Control and Measurement Systems Supporting the Production of Haylage in Baler-Wrapper Machines"

_sensors, 2023, doi:10.3390/s23062992_

Round 1

Reviewer 1 Report

1. The second and third paragraphs in the introduction are not relevant to the research topic of the paper ,and should be deleted appropriately.

2. The introduction mainly introduces some products, but lacks the introduction of relevant research methods

3.  A more detailed explanation of Figure 3 should be added. How and why did different tractor tracks affect the forces of pressing?  The installation position and working principle of the sensor should also be explained.

4. The accuracy of  the swath volume measuring system.

5. Is there are fitting formula for the relationship between PTO torque , pressing force and  weight

6.Quantitative realization of Variable dosing of ensilage additives

7.There are errors in the English expression of the paper, which need to be carefully revised.

Reviewer 2 Report

it seems to me an interesting and well done job, I have no additional request

1. What is the main question addressed by the research?  

The use of 3D sensors to optimize the operativity of baler-wrapper machines

2. Do you consider the topic original or relevant in the field? Does it

address a specific gap in the field?   It is a useful novelty in the sector, I don't know similar devices already marketed  

3. What does it add to the subject area compared with other published

material?  

See the previous answer

4. What specific improvements should the authors consider regarding the

methodology? What further controls should be considered?  

I really don’t know, in my opinion this work was carried out and subsequently written very well

  5. Are the conclusions consistent with the evidence and arguments presented and do they address the main question posed?  

Yes

6. Are the references appropriate?  

Yes

7. Please include any additional comments on the tables and figures.  

They are clear and complete (is the first time that I’ve nothing to add)

Reviewer 3 Report

The manuscript describes selected automation topics that improve efficiency, productivity, operational reliability, ergonomics from the operator's point of view, and, most importantly, control systems that affect the quality of feed for livestock feeding. To control the process of creating baler-wrappers and regulate the forage ensiling process, the manuscript presents the support system of a method for estimating the volume of swaths raked into rows. The introduction confirms that the use of modern precision farming systems allows controlling the process of creating bales and regulating the forage ensiling process, and has been proven with the example of the method of density control, the concept of dosing ensiling agents, recording the weight and volume of the swath. Suggestions for further revisions.

1.     The main novelty and contributions of this work should be better explained. It is recommended that the paper focus more on one aspect of the control to do so, rather than all aspects.

2.     The title of the manuscript does not reflect well the content covered by the paper, and it is recommended to revise it. In particular, it is only a simple control and measurement system, and the part that focuses on the baler-wrapper needs to be clarified.

3.     Update the reference list by including newly published papers.
